# LRRK2-phosphorylated Rab10 sequesters Myosin Va with RILPL2 during ciliogenesis blockade

Herschel S Dhekne[1],* , Izumi Yanatori[1],* , Edmundo G Vides[1] , Yuriko Sobu[1] , Federico Diez[2], Francesca Tonelli[2] , Suzanne R Pfeffer[1]

Activating mutations in LRRK2 kinase causes Parkinson's disease. Pathogenic LRRK2 phosphorylates a subset of Rab GTPases and blocks ciliogenesis. Thus, defining novel phospho-Rab interacting partners is critical to our understanding of the molecular basis of LRRK2 pathogenesis. RILPL2 binds with strong preference to LRRK2-phosphorylated Rab8A and Rab10. RILPL2 is a binding partner of the motor protein and Rab effector, Myosin Va. We show here that the globular tail domain of Myosin Va also contains a high affinity binding site for LRRK2-phosphorylated Rab10. In the presence of pathogenic LRRK2, RILPL2 and MyoVa relocalize to the peri-centriolar region in a phosphoRab10-dependent manner. PhosphoRab10 retains Myosin Va over pericentriolar membranes as determined by fluorescence loss in photobleaching microscopy. Without pathogenic LRRK2, RILPL2 is not essential for ciliogenesis but RILPL2 over-expression blocks ciliogenesis in RPE cells independent of tau tubulin kinase recruitment to the mother centriole. These experiments show that LRRK2 generated-phosphoRab10 dramatically redistributes a significant fraction of Myosin Va and RILPL2 to the mother centriole in a manner that likely interferes with Myosin Va's role in ciliogenesis.

## Introduction

Activating mutations in the LRRK2 kinase cause Parkinson's disease (Alessi & Sammler, 2018). Recent work has shown that pathogenic LRRK2 phosphorylates a subset of Rab GTPases, including Rabs 8, 10, 12, and 35 (Steger et al, 2016, 2017) that are master regulators of membrane trafficking events (Pfeffer, 2017). A significant consequence of LRRK2 Rab phosphorylation is a decrease in primary cilia in a variety of cell types, including in mouse brain (Steger et al, 2017; Dhekne et al, 2018; Lara Ordóñez et al, 2019). Phosphorylation of Rab proteins blocks their abilities to interact with key regulators including Rab GDI and the Rabin8 guanine nucleotide exchange factor, as well as multiple, cognate Rab effector proteins (Steger et al, 2016). This loss of effector binding alone would be sufficient to interfere with Rab protein physiological functions. Importantly, however, phosphorylated Rab proteins also show enhanced binding to novel effectors, and understanding the roles of these nascent interactions is critical to our understanding of the molecular basis of Parkinson's disease pathogenesis.

RILP is a Rab7 effector and so-called, "cargo adaptor" that links the microtubule-based motor proteins, dynein/dynactin to late endosomes (Cantalupo et al, 2001; Jordens et al, 2001). RILPL1 and RILPL2 are two RILP-related proteins that bind much more tightly to phosphorylated Rab8 and Rab10 proteins than to their non-phosphorylated forms (Steger et al, 2017), but little is known about their cellular roles. Both RILPL1 and RILPL2 were reported to be ciliary proteins involved in regulating ciliary content (Schaub & Stearns, 2013). We showed previously that RILPL1 is relocalized quantitatively to the mother centriole by (phosphorylated) pRab10 protein, and both pRab10 and RILPL1 (but not Rab8A) are essential for the ability of pathogenic LRRK2 to block cilia formation (Dhekne et al, 2018). In this study, we have investigated the possible role of RILPL2 in contributing to a LRRK2-triggered cilia blockade.

RILPL2 is a Myosin Va (MyoVa)–interacting protein first reported to be important for cell shape control and neuronal morphogenesis (Lisé et al, 2009). RILPL2 interacts with the C-terminal, globular tail domain (GTD) of MyoVa via its N-terminal RILP-homology "RH1" domain (Lisé et al, 2009) and binds pRab8 and pRab10 (Steger et al, 2017) via its C-terminal RH2 domain (129–165; cf. Waschbüsch et al, 2020). Recently, RILPL2 has also been reported to regulate the motor activity of MyoVa (Cao et al, 2019). By binding a myosin motor at one end, and a membrane-anchored Rab at the other, RILPL2 is an actin-based, motor-adaptor protein.

We have been studying the effect of Rab phosphorylation on ciliogenesis and report here our progress in understanding the consequences of pRab10-RILPL2 complex formation. We show that the MyoVa GTD also contains a high affinity binding site for LRRK2-phosphorylated Rab10, and LRRK2 kinase activity and pRab10 drive MyoVa and RILPL2 to the mother centriole during ciliogenesis blockade, retarding the release of MyoVa from that location.

[1]Department of Biochemistry, Stanford University School of Medicine, Stanford, CA, USA    [2]Medical Research Council Lab of Protein Phosphorylation and Ubiquitylation, University of Dundee, Dundee, Scotland

Correspondence: pfeffer@stanford.edu
*Herschel S Dhekne and Izumi Yanatori contributed equally to this work

Quantitative analysis of Rab10 and MyoVa copy numbers in various cell types indicates that a large pool of total MyoVa may be sequestered under these steady state conditions.

# Results

## RILPL2 relocalizes to the mother centriole in serum starved cells

RILPL2 associates with primary cilia in IMCD3 cells (Schaub & Stearns, 2013). We determined the localization of RILPL2 in primary astrocytes that were isolated using antibody immuno-panning from brains of newborn rats (Foo et al, 2011). Most of these astrocytes contain Arl13b[+] primary cilia, and endogenous RILPL2 was detected at low levels throughout the cytoplasm and also at the base of these cilia (Fig 1A). Cultured cell lines, including hTERT-(RPE) retinal pigment epithelial cells, yielded only weak endogenous RILPL2 staining; however, upon exogenous expression, HA- and GFP-RILPL2 displayed a diffuse, cytoplasmic localization (Fig S1A and below). After 2 h of serum starvation to induce primary cilia formation, a small amount of HA-RILPL2 was detected adjacent to the mother centriole, identified by distal appendage protein CEP164, or centriolar cap protein CP110 that labels both centrioles (Fig 1B); partial RILPL2 re-localization upon serum withdrawal was observed in >50% of the transfected cells (Fig 1C).

Upon live imaging of hTERT-RPE cells transfected with GFP-RILPL2 and monomeric kusabira-orange (mKO2) tagged with the centrosomal targeting sequence of Pericentrin/AKAP450 (PACT) to provide a centrosomal marker (Gillingham & Munro, 2000), RILPL2 arrived at the mKO2-PACT[+] puncta after ~3 h of serum starvation, and remained associated with the centrosome for at least the next 5 h (Fig 1D and E). Thus, serum starvation increases the pericentriolar localization of RILPL2 protein.

## LRRK2 activity relocalizes RILPL2 and MyoVa

We showed previously that LRRK2 kinase-phosphorylated Rab10 is strongly relocalized to the pericentriolar region with RILPL1 protein (Dhekne et al, 2018). Fig 2A shows an example of HeLa cells expressing pathogenic R1441G-LRRK2–generated pRab10, that is tightly concentrated on membranes surrounding Centrin 3–labeled centrioles (Fig 2A). We explored the effect of LRRK2 activity on the localization of RILPL2. In HEK293T cells expressing R1441G LRRK2 to generate pRab10, RILPL2 localized tightly to perinuclear, pRab10-positive structures (green, Fig 2B top row, Fig 2C and D), unlike the distribution seen in wild-type cells (Fig S1A).

Most MyoVa is distributed diffusely throughout the cytoplasm (cf. Fig S1A), but a small amount associates with primary cilia (Kohli et al, 2017) and regulates ciliogenesis (Assis et al, 2017). In cells expressing pathogenic LRRK2, MyoVa was redistributed to RILPL2 and pRab10-positive membranes (red, Fig 2B). Addition of the MLi-2 LRRK2 inhibitor for 2 h completely reversed the phenotype: pRab10 staining was lost and both RILPL2 and MyoVa proteins reverted to a diffuse localization in 90% of cells (Fig 2B lower row; Fig 2C). Importantly, localization of RILPL2 to the pericentriolar region did not depend on MyoVa because three different shRNAs could deplete

MyoVa effectively (Fig 2E) and in these stable cell lines, RILPL2 clustering was unaltered (Fig 2D–F). Because we were unable to detect endogenous RILPL2 in these cell types, these data reflect the behavior of exogenous HA-tagged RILPL2 protein. Similar findings were obtained for R1441C MEF cells (Fig S1B and C); HA-RILPL2 localized to numerous pericentriolar vesicles and its localization became much less concentrated in the presence of MLi-2 to inhibit LRRK2 activity. In addition, MyoVa knockdown (KD) in LRRK2-expressing HEK293T cells did not alter the localization of pRab10 (Fig S1D and E). Altogether, these experiments show that RILPL2 relocalizes to pRab10-positive, pericentriolar membranes, independent of MyoVa protein.

MyoVa also relocalizes to pericentriolar structures: although it could be carried there by RILPL2 or Rab10 proteins (Lisé et al, 2009; Roland et al, 2009), we show next that MyoVa binds directly to pRab10 and relocalizes independent of RILPL2 protein (Fig S2D–F).

## MyoVa GTD drives co-localization with MyoVa

Fig 3A diagrams the domain structure of Myosin V proteins. The N-terminal motor domain is followed by a series of isoleucine-glutamine (IQ) motifs and a set of variably spliced exons. At the C terminus, the MyoVa GTD can interact with Rab3A (Wöllert et al, 2011), Rab11 (Roland et al, 2009; Lindsay et al, 2013; Pylypenko et al, 2013, 2016), and with Rab27A via the Slac2/Melanophilin adaptor (Fukuda et al, 2002; Hume et al, 2002; Wei et al, 2013) in different cell types. Importantly, Exon D that is conserved in all Myosin V isoforms contains a binding site for Rab10 (Roland et al, 2009). Exon D-containing forms of MyoVa are expressed broadly, but not in brain or endocrine/neuroendocrine cells (Seperack et al, 1995; Roland et al, 2009). We, therefore, created a MyoVa truncation construct that deletes Exon D (MyoVa ΔD lacking amino acids 1,320–1,345) and a construct to express the GTD alone (residues 1,421–1,880; Fig 3A).

In HEK-293T cells, mCherry-tagged versions of MyoVa FL (full length MyoVa) and GTD were mostly cytosolic (Fig 3B). When MyoVa FL was co-expressed with GFP-Rab10, more of it re-localized to perinuclear, Rab10 containing vesicles (Fig 3C top row), consistent with its established binding capacity. In contrast, Exon D–deleted MyoVa (MyoVa ΔD) showed much less colocalization with GFP-Rab10 (80 versus ~30%, Fig 3C lower row, Fig 3E). Surprisingly, expression of R1441G pathogenic LRRK2 relocalized both MyoVa FL and MyoVa ΔD to perinuclear membrane structures positive for pRab10 (Figs 3D and S2A). Quantitation revealed that MyoVa colocalized with pRab10 to almost the same extent, with or without Exon D (Fig 3E). Importantly, even the normally diffuse GTD (Fig 3B) redistributed to pericentriolar, pRab10[+] membranes (Fig 3F): 35% of total MyoVa GTD colocalized with pRab10, whereas ~80% of pRab10 co-localized with GTD (Fig 3G). These values match closely the extent of RILPL2 relocalization seen in earlier experiments (Fig 2). Finally, endogenous MyoVa also co-localized with pRab10 in LRRK2 G2019S[+/−] primary astrocytes and GFP-R1441G-LRRK2 expressing HEK293T cells (Fig S3A and B).

The ability of pRab10 to relocalize both MyoVa ΔD and the GTD suggested that the MyoVa GTD might also contain a pRab10-specific binding site. Therefore, we tested the requirement for Rab10 protein to drive full-length MyoVa ΔD or GTD relocalization in R1441G LRRK2–expressing 293T cells (Fig 4). Whereas control cells showed

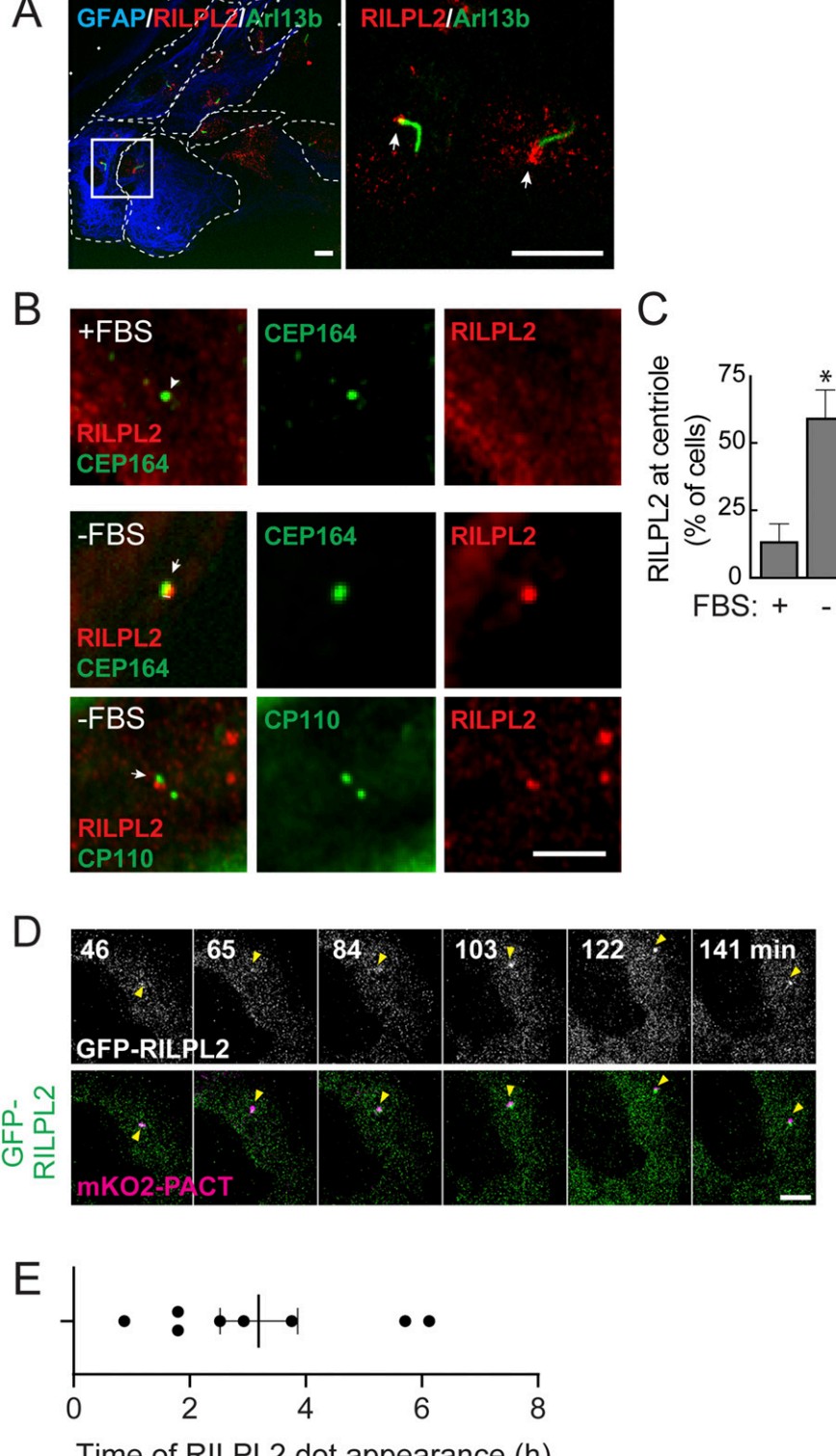

**Figure 1. RILPL2 re-localizes to the mother centriole in serum-starved cells.**
**(A)** Primary, wild type rat astrocytes were fixed with 3% PFA and stained for cilia using mouse anti-Arl13b (green) and rabbit anti-RILPL2 (red). Box indicates the region enlarged in the image shown at right. Bar, 10 μm. Arrows indicate the base of the cilium. **(B)** hTERT-RPE cells were transfected with HA-RILPL2 at 80% confluency and 24 h later either medium changed (+FBS) or serum starved (−FBS) for 2 h. Cells were fixed with −20°C methanol for 2 min and stained with rabbit or mouse anti-HA to detect RILPL2 (red) and mouse anti-CEP164, or rabbit anti-CP110 (green). Arrows indicate the location of the centriolar region. Bar, 2.5 μm. **(C)** Quantitation of percent of cells that show centriolar localized RILPL2 ± 2 h serum starvation. Significance was determined by *t* test; *P = 0.03. Error bars represent standard error of the mean from two independent experiments with >50 cells per condition. **(D)** hTERT-RPE cells expressing mKO2-PACT (magenta) were transfected with GFP-RILPL2 (green). Cells were serum starved and 15 min later imaged by capturing Z-stacks every 6 min for the next 8 h. Yellow arrowheads indicate the location of the centrosome as marked by mKO2-PACT. Upper panel shows GFP-RILPL2 in grayscale. Scale bar, 5 μm. **(E)** Times at which GFP-RILPL2 appeared at the mKO2-PACT structure. Error bar represents standard error of the mean.

tight perinuclear GTD staining that was coincident with pRab10, cells treated with two different Rab10 shRNAs showed no pRab10 staining and a diffuse GTD localization (Fig 4A, C, and E). Rescue with Myc-Rab10 protein restored the original, concentrated GTD localization (Fig 4B, D, and E). Rab10 depleted cells also showed dispersed full length MyoVa mCherry (Fig S2A) as well as dispersed GFP-RILPL2 (Fig S2C). Thus, the GTD requires Rab10 protein for pericentriolar localization; it appears not to require RILPL2 as RILPL2 depletion in R1441C MEF cells did not alter GTD localization (Fig S2D–F).

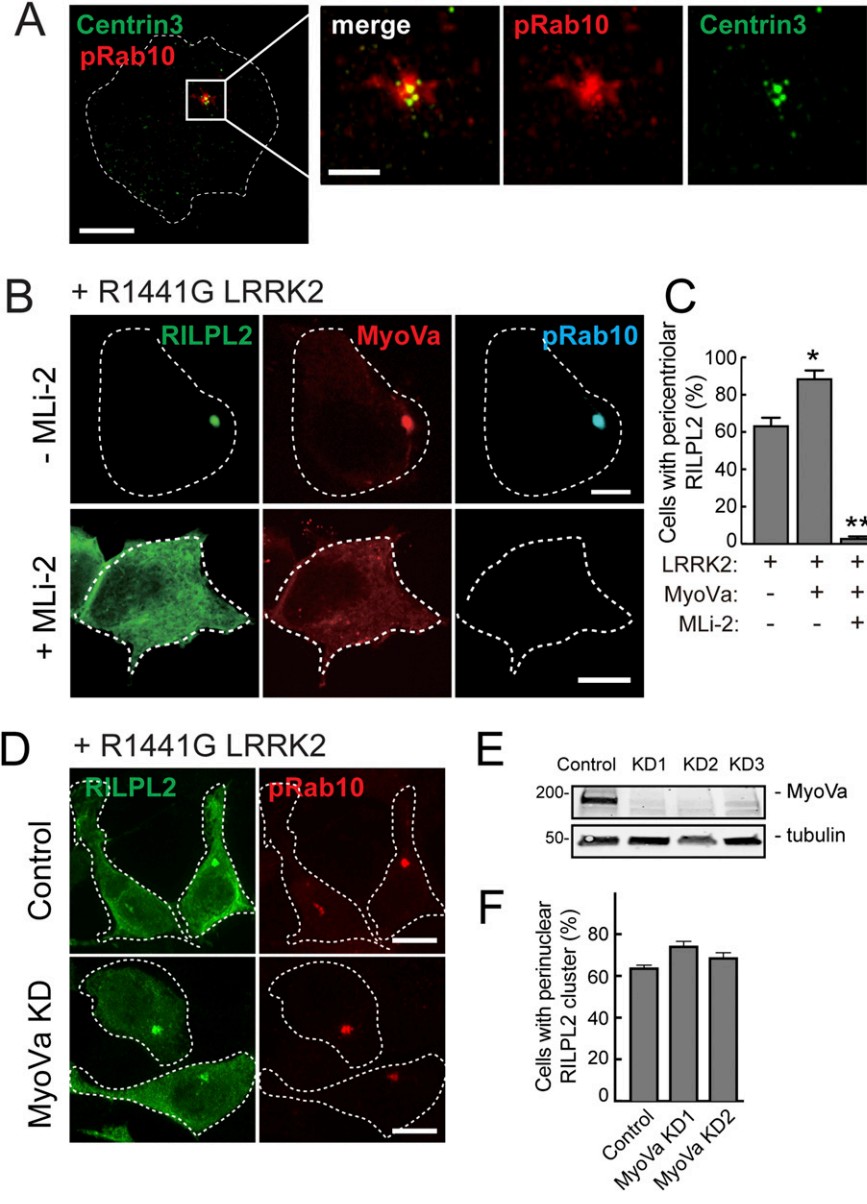

**Figure 2. PhosphoRab10 drives centriolar relocalization of MyoVa and RILPL2.**
**(A)** Localization of pRab10 and the centriole marker, Centrin 3. HeLa cells were transfected with R1441G Myc-LRRK2 and methanol fixed for staining with rabbit anti-pRab10 and mouse anti-Centrin 3. The boxed area is enlarged in the adjacent panels at right. Bar, 10 μM.
**(B)** HEK293T cells were transfected with R1441G LRRK2, full-length MyoVa mCherry, and HA-RILPL2 for 16 h, treated ± 200 nM MLi-2 for 2 h and stained with mouse anti-HA and rabbit anti-phosphoRab10. Bar, 10 μm. **(C)** Quantitation of cells with perinuclear RILPL2 from the experiment in (B) from two independent experiments with >50 cells per condition. *0.027; **0.0026 *t* test. **(B, D)** HEK293T cells depleted of MyoVa were transfected with R1441G LRRK2 and HA-RILPL2 and stained for RILPL2 and pRab10 as in (B).
**(E)** Immunoblot of HEK293T cells depleted of MyoVa using three shRNAs to create stable cell lines. Numbers at left in this and subsequent figures represent mass in kilodaltons. **(F)** Quantitation of perinuclear RILPL2 in cells with and without MyoVa. No significant difference was detected.

## Purified MyoVa GTD binds phosphorylated Rab10 directly

Co-immuno-precipitation experiments in cells expressing R1441G LRRK2 confirmed binding of the MyoVa GTD to pRab10. As shown in Fig 5A, full-length MyoVa mCherry bound both Rab10 and pRab10; binding of both forms decreased in the presence of MLi-2 LRRK2 inhibitor, suggesting that pRab10 was a significant contributor to MyoVa binding. Similarly, MyoVa ΔD lacking the D exon–Rab10 binding site still bound roughly the same amount of pRab10 and less total Rab10 protein (Fig 5A and B). These experiments strongly suggest that MyoVa ΔD contains an additional binding site with preference for pRab10. Indeed, similar analysis of binding to the GTD domain in cell extracts (Fig 5C) confirmed co-immunoprecipitation of Rab10 and pRab10 binding to the GTD, both of which were completely lost upon LRRK2 inhibition.

In experiments in which we monitored binding to RILPL2-GFP, MyoVa GTD bound with or without LRRK2 activity, but we detected about twice as much MyoVa GTD and Rab10 precipitating with RILPL2 in cells expressing R1441G LRRK2 (Fig 5D and E). These data are consistent with pRab10 binding both RILPL2 and MyoVa. Because RILPL2 can dimerize (Wei et al, 2013; Waschbüsch et al, 2020), it is possible that each half of the dimer engages either MyoVa or pRab10 or both. Alternatively, pRab10 may enhance the ability of MyoVa to bind RILPL2. Because pRab10 re-localization of MyoVa and RILPL2 could occur in the absence of the other partner, these possibilities were not studied further.

To demonstrate more directly the presence of a pRab10-specific binding site in the MyoVa GTD, we used microscale thermophoresis (MST) to monitor binding of purified proteins. Bacterially expressed, purified Rab10 was phosphorylated in vitro using recombinant Mst3, a kinase used previously to efficiently phosphorylate purified Rab8

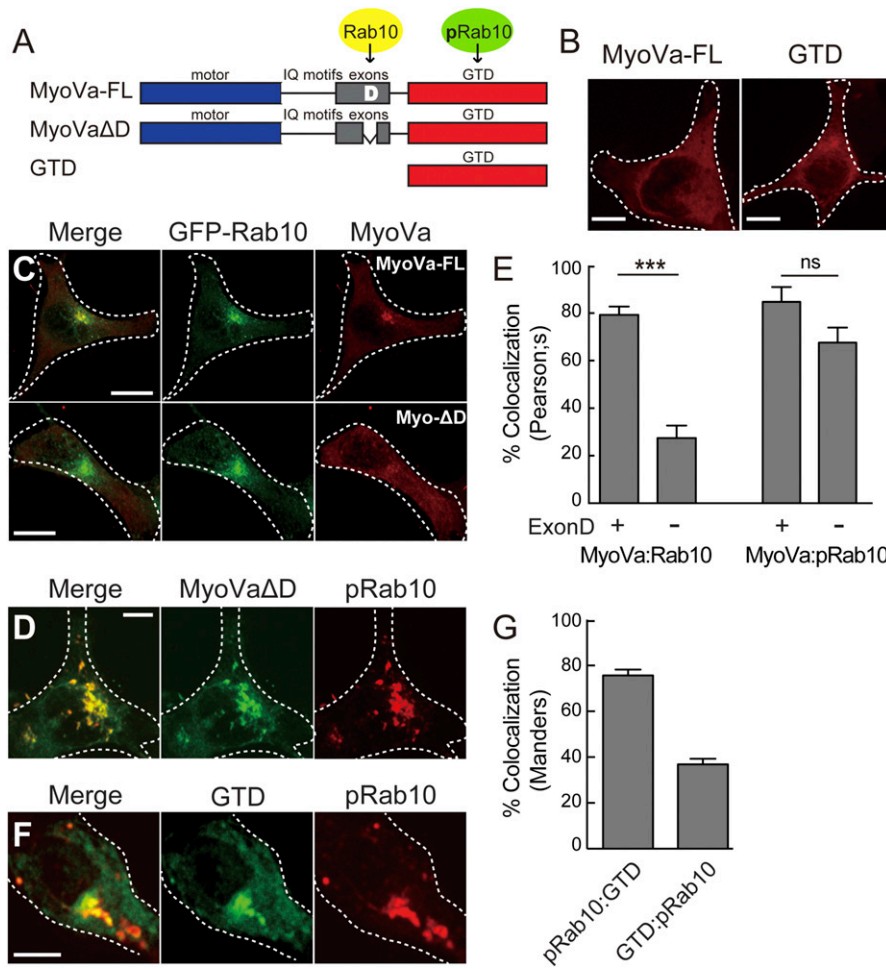

**Figure 3. MyoVa exon D and its globular tail domain drive Rab10 colocalization.**
**(A)** MyoVa domain structure. MyoVa has a motor domain (blue), isoleucine–glutamine (IQ) motifs, five exons (A, B, C, D, E, gray) and a globular tail domain (GTD, red). MyoVa ΔD lacks exon D and its Rab10 binding site; GTD indicates a construct that contains only the MyoVa C-terminus (and pRab10 binding site). **(B)** Full-length MyoVa mCherry (MyoVa FL) or GTD-mCherry were transfected into HEK293T cells plated on collagen coated coverslips and imaged after 24 h. Bars, 10 μm. **(C)** GFP-Rab10 was co-transfected with full length MyoVa mCherry (top row) or Exon D deleted MyoVa mCherry (bottom row) in HEK293T cells and fixed after 24 h. Bar, 10 μm. **(D)** Myc-LRRK2 R1441G was co-transfected with Exon D deleted-MyoVa mCherry in HEK293T cells and fixed after 24 h. Cells were stained for pRab10 (red), whereas MyoVa mCherry is pseudo-colored (green). Bar, 10 μm. **(E, left)** Quantitation of colocalization by Pearson's method between GFP-Rab10 and MyoVa mCherry ± Exon D (as indicated). **(E, Right)** Quantitation of colocalization by Pearson's method between pRab10 and MyoVa mCherry ± Exon D (as indicated). **(F)** Myc-LRRK2 R1441G was co-transfected with GTD-mCherry in HEK293T cells and fixed after 24 h. Cells were stained for pRab10 (red), whereas GTD-mCherry is pseudo-colored (green). Bar, 10 μm. **(G)** Quantitation of colocalization by Mander's co-occurrence method where fraction pRab10 that colocalizes with GTD is pRab10:GTD and fraction of GTD that colocalizes with pRab10 is GTD:pRab10. Scale bars, 10 μm. Error bars indicate SEM from three independent experiments with >50 cells per condition. Significance was determined by $t$ test; ***$P$ = 0.0001; ns, not significant with $P$ = 0.0673.

and Rab10 proteins, uniquely at the same site as LRRK2 (Berndsen et al, 2019; Waschbüsch et al, 2020). In vitro phosphorylation of Rab10 proceeded linearly for ~100 min before reaching a plateau in reactions containing 2 mM ATP and a ratio of 1:3 kinase:substrate at 27°C, as detected using a phosphoRab10-specific antibody (Fig 6A). As expected, the reaction was ATP dependent (Fig 6B), and phosphorylation was efficient: ~90% of Rab10 was phosphorylated in reactions containing Mst3 kinase as determined by PhosTag gel electrophoresis that enables resolution of phosphorylated and non-phosphorylated Rab10 proteins (Fig 6C).

Because it is difficult to purify significant quantities of fully active and phosphorylated Rab10, we established a protocol that would enable us to monitor in vitro pRab10 binding to MyoVa with minimal protein handling. As outlined in Fig 6D, Rab10 was incubated for 120 min with Mst3 kinase and ATP to achieve phosphorylation. NT-647 dye–labelled GTD was added for a subsequent 30-min binding reaction, followed by transfer of the binding mix into capillaries for microscale thermophoresis analysis. Control reactions contained all the same components except ATP during binding; no signal was observed in reactions lacking Rab10 protein.

Phosphorylated Rab10 bound to purified MyoVa GTD with a $K_D$ of 475 ± 166 nM under these conditions (Fig 7A). This represents strong binding as most Rab effectors bind Rabs with low micromolar affinities. Non-phosphorylated Rab10 assayed in parallel showed no

binding to GTD in reactions containing up to 18 μM GTD (Fig 7B). For comparison, we also tested binding to another LRRK2 substrate, Rab8A. The MyoVa GTD bound pRab8 ~14-fold less strongly and no binding was detected to non-phosphorylated Rab8A protein (Fig 7C and D). In summary, these in vitro binding assays confirm that pRab10 binds directly and with high affinity to the MyoVa GTD; pRab8A binds much more weakly, consistent with the requirement for Rab10 protein in cells for MyoVa relocalization, despite the presence of endogenous Rab8A protein (Fig 4).

## Exogenous RILPL2 localizes to the mother centriole and blocks ciliogenesis

We showed previously that overexpression of RILPL1 was sufficient to block ciliogenesis in RPE and A549 cells (Dhekne et al, 2018). In addition, LRRK2 generation of pRab10 was sufficient to block ciliogenesis but this required that cells contain endogenous RILPL1 protein (Dhekne et al, 2018). We thus tested the effect of exogenous RILPL2 expression on primary cilia formation in the absence of pathogenic LRRK2. hTERT-RPE cells were transfected with HA-tagged RILPL2 and 24 h later, serum starved overnight to initiate ciliogenesis. As shown in Fig 8A and B, even in the absence of pathogenic LRRK2, RILPL2 expression strongly inhibited cilia formation. In addition, clonal

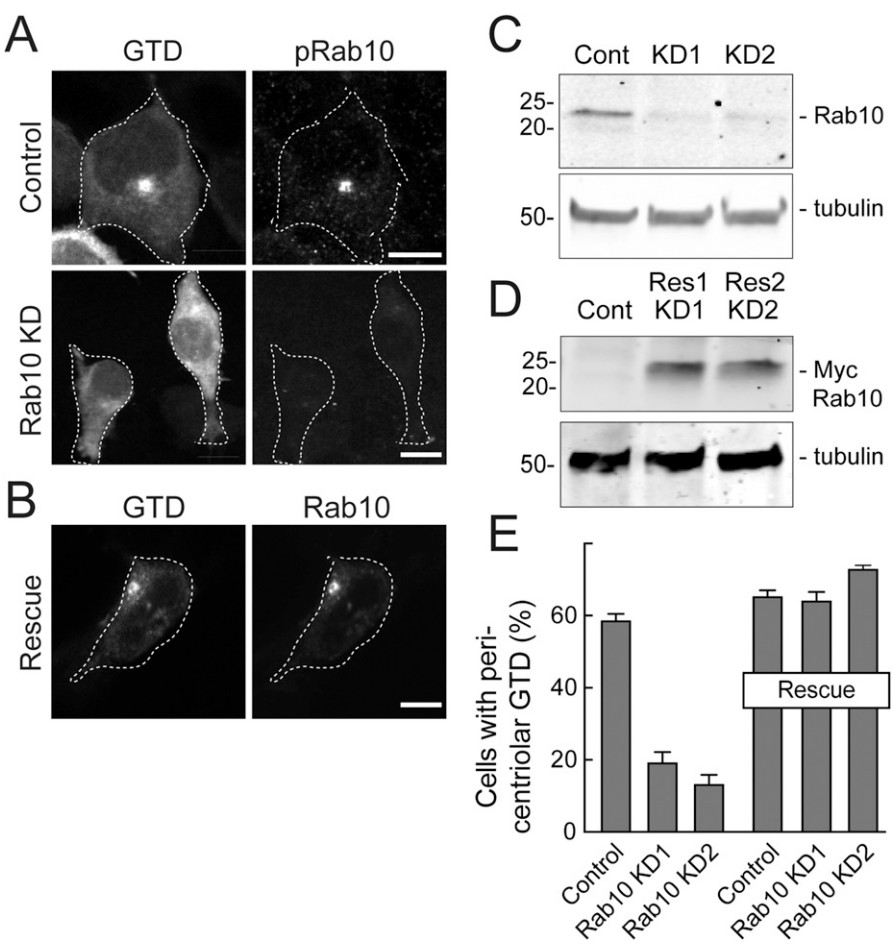

**Figure 4. Rab10 is essential for MyoVa globular tail domain (GTD) relocalization.**
**(A)** HEK293T cells depleted of Rab10 using two shRNAs (KD1 and KD2) were transfected with MyoVa GTD and R1441G LRRK2 for 16 h. Lentivirus carrying scrambled shRNA-treated cells were used as controls. MyoVa GTD and pRab10 were visualized as in Fig 2. **(B)** Cells were transfected with Myc-Rab10 for 16 h and stained as in (A). **(C, D)** Immunoblot analysis of Rab10, Myc-Rab10, and tubulin loading control as in Fig 2. Rescue 1 (Res1) and rescue 2 (Res2) refer to the different rescue plasmids corresponding to the distinct shRNAs stably expressed. **(E)** Quantitation of cells with pericentriolar GTD without (left bars) or with (right bars) Myc-Rab10 rescue as indicated. Shown are the data from two experiments (>200 cells per experiment).

RILPL1 and RILPL2 knockout A549 cell lines were created using CRISPR-Cas9. As we reported previously (Dhekne et al, 2018), endogenous RILPL1 depletion in these cells dramatically increased cilia formation, consistent with RILPL1's role as a suppressor of ciliogenesis. In contrast, RILPL2 depletion did not influence ciliation in two independent knockout lines (Fig 8C). Consistent with these results, MEFs generated from homozygous RILPL2 knock-out embryos also showed no changes in overall ciliogenesis (Fig 8B, right panel). Nevertheless, exogenous expression of RILPL2 in these knockout MEF cells inhibited cilia formation to an extent similar to that seen in RPE cells (Fig 8B).

Ciliogenesis is initiated by Tau tubulin kinase 2 (TTBK2)–mediated phosphorylation events that drive the uncapping of CP110 from the mother centriole (Spektor et al, 2007; Goetz et al, 2012). CP110 uncapping also requires EHD1-dependent ciliary vesicle formation, a MyoVa catalyzed process (Lu et al, 2015; Wu et al, 2018). We found that CP110 release required MyoVa in RPE cells under the conditions of our experiments (Figs 8D and S4A). We also found that RILPL2 overexpression blocked CP110 release (Fig 8D) but not TTBK2 recruitment to the mother centriole (Figs 8E and S4A and B). Moreover, HA-RILPL2 expression decreased the overall concentration of MyoVa at the centriole after 2 h of serum starvation (Figs 8F and S4C). These data indicate that overexpression of RILPL2 in some way interferes with the ability (or availability) of MyoVa to support the ciliary vesicle recruitment needed for CP110 uncapping and subsequent ciliogenesis.

Wu et al (2018) reported that MyoVa knockout cells showed no effect on CP110 release but had a striking defect in ciliary vesicle association with mother centriole. This difference might be explained by adaptation in knockout clones and/or a very short time of serum starvation (30 min) in Wu et al (2018) (Fig S4B) versus acute, siRNA-mediated MyoVa depletion and longer serum starvation (2 h in serum free medium) in our experiments (Fig 8D).

It was not possible to test if RILPL2 expression blocks ciliogenesis in a MyoVa dependent manner because MyoVa knockdown alone led to a block in CP110 capping (Fig 8D). Nevertheless, these experiments suggest that exogenous overexpression of HA-RILPL2 blocks ciliogenesis by blocking MyoVa's ability to participate in ciliogenesis initiation.

## Pericentriolar retention of MyoVa by pRab10

The strong concentration of pRab10, RILPL2, and MyoVa over the centriole in pathogenic LRRK2-expressing cells suggested that MyoVa may become sequestered by pRab10 and hindered from contributing to ciliogenesis initiation. To explore this, we turned to fluorescence loss in photobleaching (FLIP), a method that decreases the fluorescence of cytoplasmic proteins to permit an estimation of live cell off-rates for proteins bound to a particular compartment (Wüstner et al, 2012). As summarized in Fig 9A, live

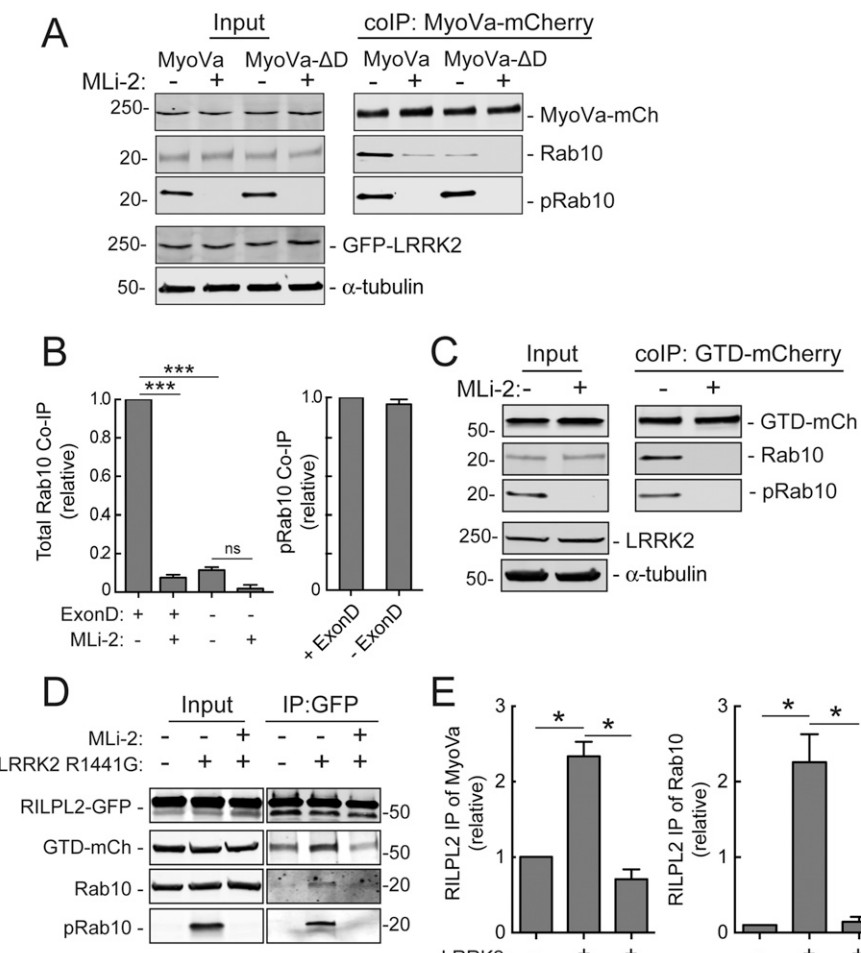

**Figure 5. pRab10-specific interaction with the MyoVa globular tail domain.**
**(A)** MyoVa or MyoVa ΔD-mCherry and GFP-LRRK2-R1441G were co-transfected into HEK293T cells; 24 h post transfection, cells were incubated with 200 nM MLi-2 for 4 h. Cells were lysed in lysis buffer and 400 μg extract immunoprecipitated with anti-RFP antibodies on protein G beads. Samples (50% of IP and 15% of input for all IPs in all panels) were immunoblotted using anti-GFP, anti-RFP, anti-tubulin, anti-Rab10 and anti-pRab10 antibodies. **(A, B)** Quantitation of the relative amount of Rab10 (left) or pRab10 (right) co-immunoprecipitated by MyoVa and MyoVa ΔD in (A). Error bars indicate standard error of the mean from two gels per co-IP. Significance was determined by $t$ test; ***$P$ = 0.0003; ns, not significant with $P$ = 0.295. **(C)** MyoVa globular tail domain (GTD)-mCherry and GFP-LRRK2 R1441G were co-transfected into HEK293T cells. After 24 h, cells were incubated ± 200 nM MLi-2 for 4 h. Cells were lysed in lysis buffer and immunoprecipitated with anti-RFP antibodies on protein G beads. Samples were immunoblotted with anti-GFP, anti-RFP, anti-Rab10 and anti-pRab10 antibodies. **(D)** RILPL2-GFP, MyoVa GTD-mCherry, and Myc-LRRK2 R1441G were co-transfected into HEK293T cells; 24 h post-transfection, cells were incubated ± 200 nM MLi-2 for 4 h. Extracts in lysis buffer were immunoprecipitated with GFP-binding protein Sepharose. Samples were immunoblotted with anti-GFP, anti-RFP, anti-Rab10 and anti-pRab10 antibodies. **(D, E)** Quantitation of the relative amount of total MyoVa GTD (left) and Rab10 (right) co-immunoprecipitated by RILPL2-GFP under the conditions indicated below (E). Significance was determined by $t$ test; *$P$ = 0.021 and 0.035 (left) and $P$ = 0.014 and 0.024 (right).

cells expressing mCherry-MyoVa constructs are bleached repeatedly (300 times every 2 s) over a small region of the cell (boxed) distinct from the perinuclear region of interest (ROI shown in dark red). Under these conditions, the cytoplasmic concentration of the labeled protein is unchanged but the off rate from an ROI can be determined over time. A fluorescence stability reference is taken from an adjacent (non-bleached) cell; fluorescence over the bleached region falls fastest, followed by the adjacent cytoplasm and last by the ROI.

Using this method in HeLa cells expressing full-length MyoVa mCherry and R1441G LRRK2, fluorescence loss over the ROI was significantly slower ($t_{1/2}$ = 272 s) than that over an adjacent area of cytoplasm ($t_{1/2}$ = 83 s) (Fig 9B–D). Importantly, this apparent retention of MyoVa mCherry was ameliorated by addition of LRRK2 inhibitor MLi-2, a condition in which the MyoVa displayed the same behavior as MyoVa in the adjacent cytoplasm (Fig 9B–D and Videos 1 and 2). Similar data and kinase activity dependence were obtained for the MyoVa GTD, although the absolute retention was decreased ($t_{1/2}$ = 107 s versus 60 s; Videos 3 and 4), possibly because GTD is monomeric whereas full length MyoVa is dimeric and avidity may contribute to longer association with pRab10-decorated vesicles.

Mutational analysis of GTD binding to pRab10 showed that a GTD R1528A mutation abolished co-localization with pRab10 in HEK293T cells

expressing R1441G LRRK2 protein; K1755A/K1757A or R1539/R1543A proteins were unaltered in their localizations (Fig 10A and B). FLIP experiments of cells expressing full-length MyoVa R1528A-mCherry showed that only the wild-type protein was capable of the pericentriolar retention detected by FLIP (Fig 10C and Video 5). The GTD contains multiple protein interaction interfaces; R1528 is important for GTD binding to melanophilin (Pylypenko et al, 2013; Wei et al, 2013). In summary, these experiments show that residue R1528 of the MyoVa GTD is critical for the ability of MyoVa to be transiently retained at the mother centriole by pRab10 protein. As discussed below, it is likely that this mode of retention interferes with the number of MyoVa molecules available to drive normal ciliogenesis initiation.

## Discussion

RILPL2 binds LRRK2-phosphorylated Rab10 with greater affinity than non-phosphorylated Rab10 (Steger et al, 2017), and although RILPL2 is known to bind MyoVa (Lisé et al, 2009) and contribute in some way to cargo enrichment in ciliogenesis (Schaub & Stearns, 2013), little is known about its precise cellular function. We have

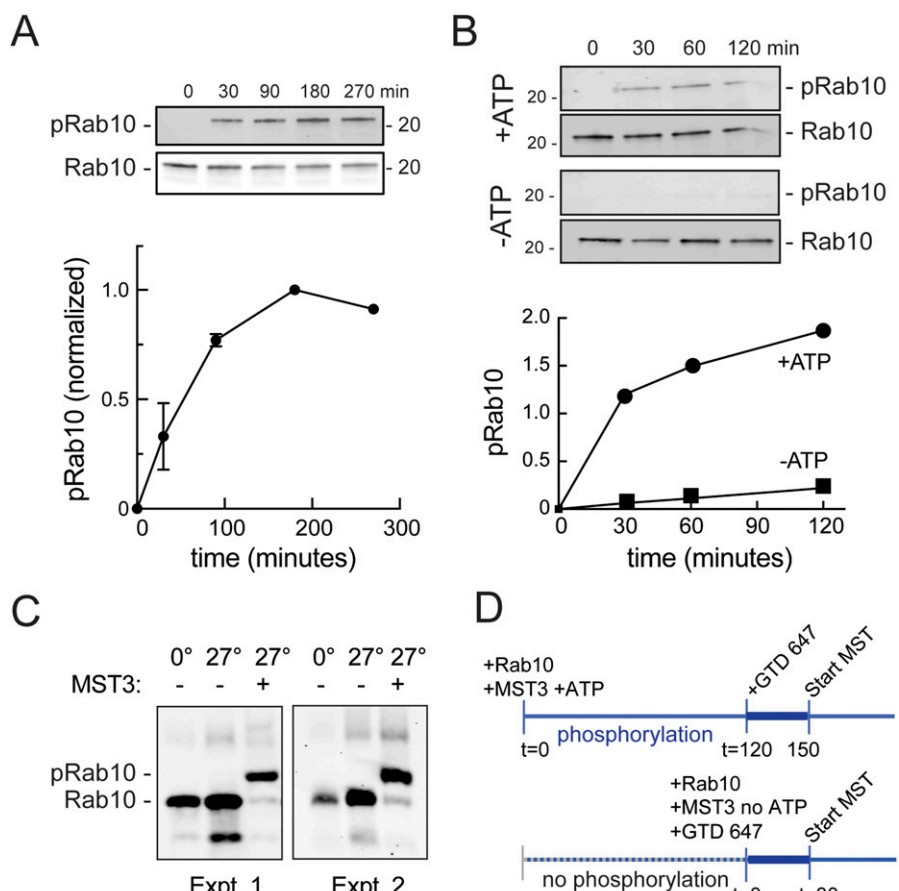

**Figure 6. In vitro phosphorylation of Rab10 by Mst3 kinase.**
**(A)** Immunoblot of reactions containing purified Rab10 and Mst3 kinase, incubated together at 27°C with 2 mM ATP. Reactions were stopped at indicated time points by addition of 2× sample buffer and boiling for 5 min. The reaction mixture (~150 ng of Rab10) was resolved by 12% SDS–PAGE. Samples were detected with anti-Rab10 and anti-pRab10 antibodies as indicated. Bottom, quantification of pRab10/Rab10 signal. **(A, B)** Purified Rab10 Q63L and Mst3 were incubated ± 2 mM ATP as in (A). **(C)** Purified Rab10-Q63L was mixed with purified Mst3 in reaction buffer as indicated and phosphorylated at 27°C for 2 h. The reaction mixture (~200–250 ng of Rab10) was resolved on a Phos-tag gel (10% gel with 40 μM Phos-tag, 80 μM MnCl$_2$), transferred onto a nitrocellulose filter after EDTA washes and probed with anti-Rab10 antibody. Two separate Phos-tag gel analyses are shown. **(D)** Schematic of Rab10 phosphorylation, binding and microscale thermophoresis experiments. In +ATP condition, Rab10 and Mst3 are incubated for 2 h at 27°C. Pre-spun NHS-Red Labeled globular tail domain (100 nM final) is then added to serially diluted pRab10 reaction mixture and incubated for 30 min in the dark at 25°C before microscale thermophoresis is started (top). In -ATP condition, the reaction mixture of Rab10 and Mst3 are serially diluted, and NHS-red labeled globular tail domain (100 nM final) is added immediately. The samples are incubated for 30 min in the dark at 25°C before microscale thermophoresis measurements are started. In all phosphorylation reactions, final concentrations of Rab10 and Mst3 were 20 and 7 μM, respectively.

shown here that RILPL2 is diffusely localized in RPE cells, and a small portion moves to the mother centriole within ~3 h of serum starvation. Others have shown that fluorescently tagged MyoVa arrives within 30 min (Wu et al, 2018), thus a small fraction of RILPL2 follows with slower kinetics.

When expression of pathogenic R1441G/C LRRK2 increases pRab10 on pericentriolar membranes, RILPL2 levels increase there dramatically. MyoVa is normally broadly distributed throughout the cytoplasm but it also concentrates in the pericentriolar region when pRab10 levels increase. Concentration of MyoVa and RILPL2 both require Rab10 protein and can occur independent of one another in cells expressing pathogenic LRRK2 kinase.

Our studies reveal that pRab10 interacts directly and tightly with MyoVa's GTD, and this interaction is likely responsible for the relocalization of MyoVa to the centriole in R1441G LRRK2-expressing cells. This was unexpected, because pRab10 could have easily influenced MyoVa localization indirectly via RILPL2 binding. However, recent mass spectroscopy analysis shows that there is most likely not adequate RILPL2 levels in cells to drive the levels of MyoVa relocalization we detect: MEFs and 293T cells contain 4,777 and 13,065 copies of RILPL2, respectively, and as much as more than 30 times more MyoVa protein (161,976 and 125,128 copies; Nirujogi et al, 2021).

The GTD of MyoVa is a versatile cargo adaptor with independent binding sites for multiple partners including RILPL2, Rab11A, Rab3A, Spire, and melanophilin. We showed that MyoVa's R1528 (conserved across tissue-specific isoforms), is critical for pRab10-dependent

relocalization. This interaction site is critical for MyoVa's binding to melanophilin in the specialized cell types that express melanophilin (Wei et al, 2013; Pylypenko et al, 2013, 2016). Indeed, mutations very close to this site have been shown to cause neurological deficits in mouse models (Huang et al, 1998). That MyoVa is also a Rab11 effector suggests that it functions as part of the Rab11:Rab8 cascade that is an important part of the early steps of cilia formation (Knödler et al, 2010; Franco et al, 2014). Note that the Rab11 interface is not predicted to be altered by the R1528 mutation (Wei et al, 2013; Pylypenko et al, 2013, 2016).

Our experiments show that MyoVa lacking exon D binds only to the phosphorylated form of Rab10 protein, much less well to phosphoRab8A, and not to non-phosphorylated Rab8. Despite its ability to bind RILPL2, endogenous Rab8 which is also phosphorylated by R1441G/C LRRK2 and located at the centriole (Purlyte et al, 2018; Lara Ordóñez et al, 2019) was not sufficient to drive RILPL2 relocalization in cells depleted of Rab10. This may be because MyoVa binds with preference to pRab10 and much less tightly to pRab8 (this work and Waschbüsch et al, 2020).

Because expression of MyoVa lacking exons D and F is specific to endocrine, neuroendocrine and neuronal cells (Seperack et al, 1995), Rab10 phosphorylation becomes a primary mode of regulation for MyoVa in these cell types. Thus, the RH homology domain proteins RILPL1, RILPL2, JIP3 and JIP4 increase their binding to Rab10 upon LRRK2 phosphorylation, whereas MyoVa lacking exon D engages a new partner in the brain and endocrine cells. This may also have cell type-specific implications for MyoVa motor activity (Cao et al, 2019).

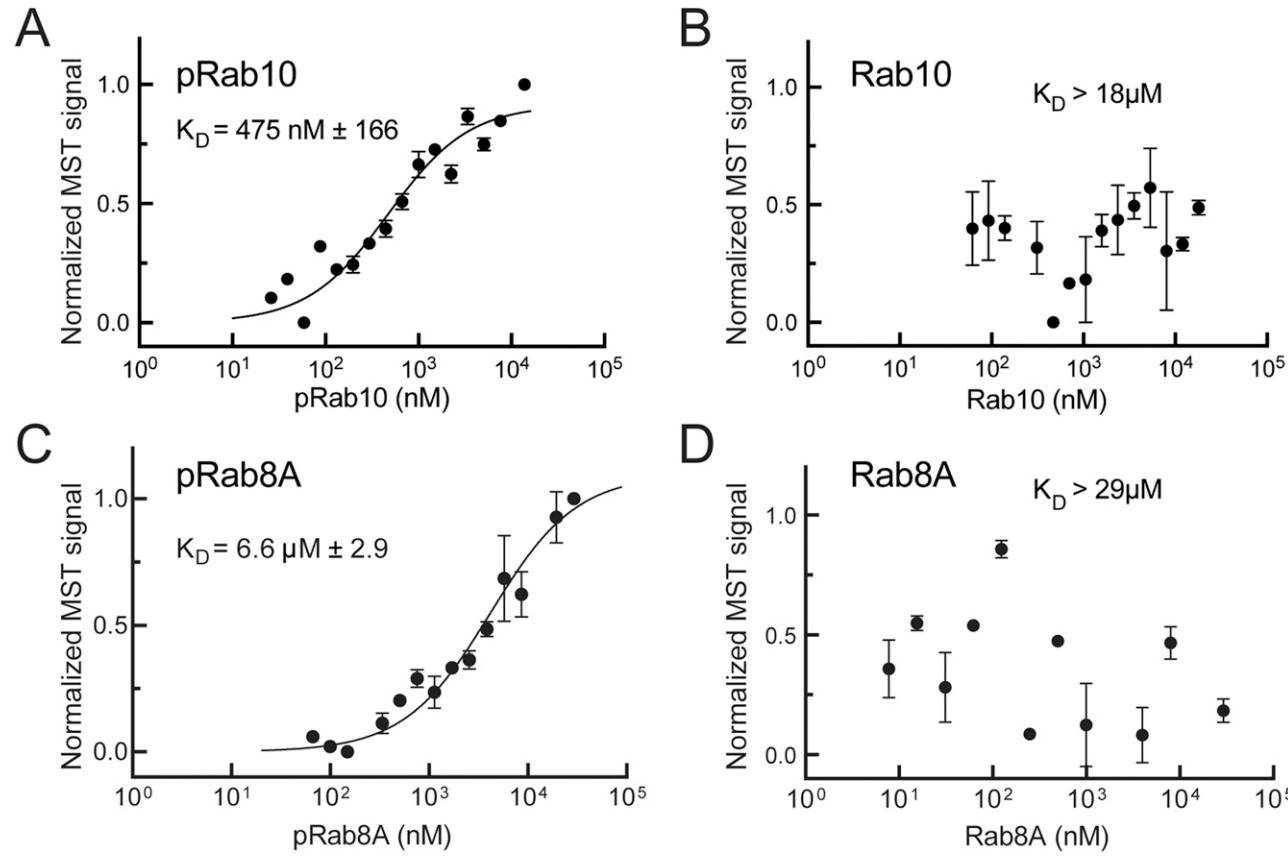

**Figure 7. pRab10 binds the MyoVa globular tail domain.**
**(A, C)** Microscale thermophoresis (MST) of the interaction between labeled MyoVa globular tail domain (GTD) with phosphorylated Rab10 or Rab8A. Purified Rab proteins were phosphorylated with Mst3 kinase at 27°C for 2 h and then serially diluted; NHS-RED labeled GTD (final concentration 100 nM) was then added. **(B, D)** MST of MyoVa GTD binding to unphosphorylated Rab10 or Rab8A. Graphs show mean and SEM from two independent measurements, each from a different set of protein preparations.

Unlike loss of RILPL1 (Dhekne et al, 2018), knockout of RILPL2 does not alter the ability of cells to form primary cilia in MEF or A549 cells. Nevertheless, like RILPL1, expression of HA-RILPL2 blocks ciliogenesis, perhaps by sequestering important components (such as MyoVa and/or Rab10) in a manner that interferes with their ability to function. Whereas RILPL1 overexpression blocks cilia formation by interfering with TTBK2 recruitment to the mother centriole (Sobu et al, 2021), RILPL2 over-expression blocks cilia formation by a different mechanism, as centriolar TTBK2 levels were not changed upon RILPL2 overexpression. It seems likely that in wild-type cells, RILPL2 overexpression interferes with MyoVa's ability to support early events in ciliogenesis that trigger CP110 uncapping independent of TTBK2.

Exogenous expression of pathogenic LRRK2 in 293T cells yields a large fraction of total Rab10 phosphorylation (Ito et al, 2016; Karayel et al, 2020). However, ciliogenesis defects are also detected in knock-in MEF cells (cf. Dhekne et al, 2018) that contain only a few percent of total Rab10 protein phosphorylation, and in MEF cells depleted of the PPM1H phosphatase that reverses LRRK2 phosphorylation (Berndsen et al, 2019). Low steady-state Rab10 phosphorylation is also seen in cells derived from patients carrying pathogenic LRRK2 mutations (Ito et al, 2016; Karayel et al, 2020).

Our FLIP experiments show strong retention of MyoVa over pericentriolar membranes and only monitor protein dissociation; reassociation will also occur, yielding the high, steady state pericentriolar concentration of MyoVa protein we observe. Note that HeLa cells contain 45,719 copies of

MyoVa and $1.9 \times 10^6$ copies of Rab10 (Itzhak et al, 2016); similarly, MEFs contain 10-fold less MyoVa than Rab10 (Nirujogi et al, 2021). Thus, even a small percentage of total Rab10 phosphorylation could capture a significant proportion of MyoVa protein. Such quantitative considerations suggest that retention of proteins such as MyoVa by pRab10-decorated vesicles can have profound consequences on the availability of proteins to contribute to specific cellular processes. Altogether, these experiments highlight the consequences of LRRK2-mediated Rab phosphorylation: a single post-translational modification causes significant subcellular redistribution and functional modulation in cells.

# Materials and Methods

### Reagents

MLi-2 LRRK2 inhibitor was obtained from Tocris Biosciences (Cat. no. 5756).

### General cloning and plasmids

DNA constructs were amplified in *Escherichia coli* DH5α or stbl3 and purified using mini prep columns (Econospin). DNA sequence

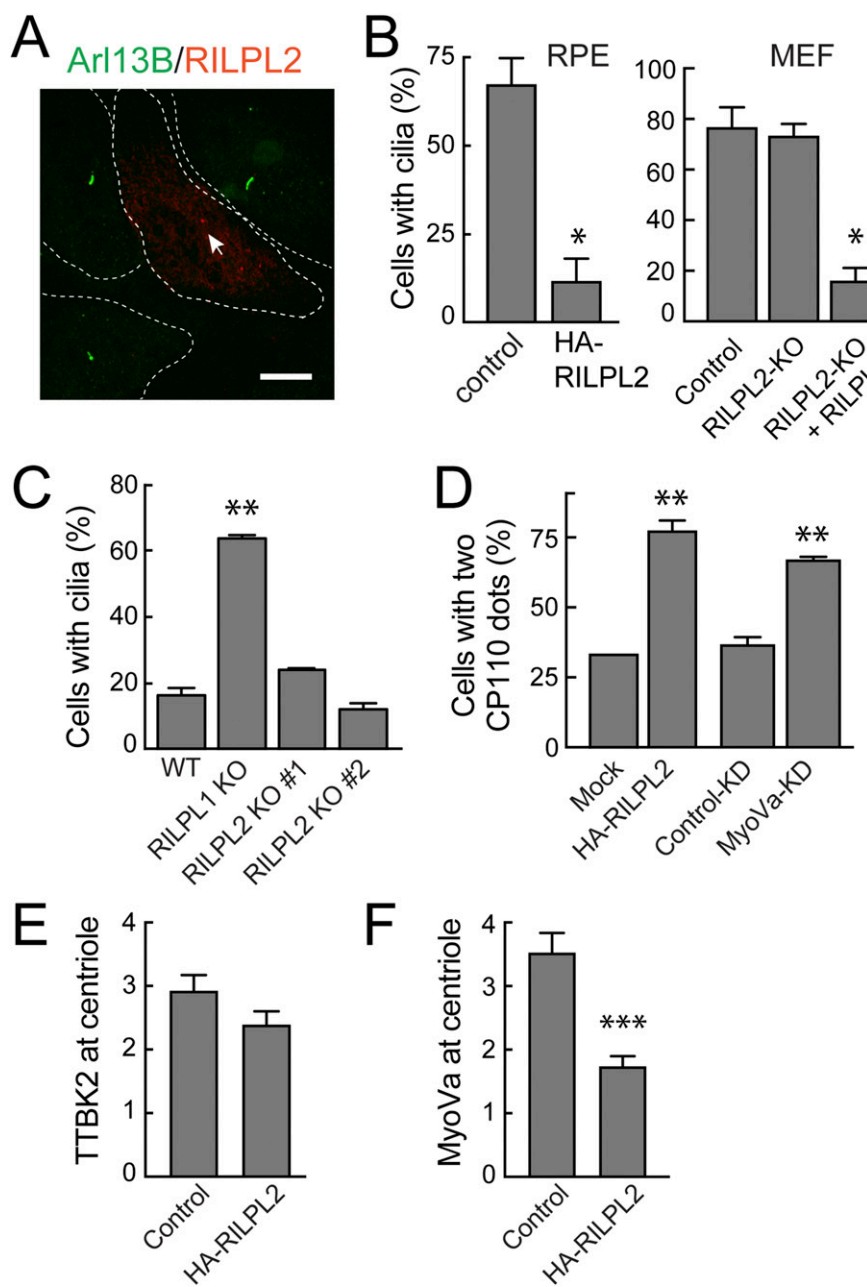

**Figure 8. Exogenous RILPL2 blocks ciliogenesis.**
**(A)** hTERT-RPE cells were transfected with HA-RILPL2 or mock transfected with lipofectamine and 24 h later, serum-starved overnight to initiate ciliogenesis. Cells were fixed with ice cold −20°C methanol for 2 min and co-stained with rabbit anti-HA (green) or mouse anti-Arl13b (red). Arrowhead indicates the likely location of a RILPL2-positive centriole. Dotted lines indicate cell outlines. Bar, 10 μm. **(B)** Quantitation of cells with cilia ± HA-RILPL2 transfection. Error bars represent SEM from two experiments, each with >25 cells per condition. Significance was determined by *t* test; *P = 0.0164. **(C)** WT, RILPL1, and RILPL2 knockout A549 cells were plated at 80% confluency. After 24 h, cells were serum starved by transfer into 2% FBS-containing medium for 48 h. Cells were fixed and stained with mouse anti-Arl13b antibody for quantitation of cells with cilia. Error bars represent SEM from two experiments with >100 cells per condition in each experiment. #1 and #2 indicate two different A549 RILPL2 knockout clonal cell lines. **(D)** Quantitation of cells with two CP110 dots indicating capped centrioles in: (left) hTERT-RPE cells ± HA-RILPL2 for 24 h; (right) hTERT-RPE cells ± MyoVa knock-down. Cells were serum starved for 2 h, fixed with −20°C methanol for 2 min and co-stained with anti-HA, anti-CP110 or anti-CEP164/CP110 antibodies. Error bars represent SEM from two experiments with >25 cells per condition in each experiment. Significance was determined by *t* test; **P = 0.0067 and 0.0045. **(B, E)** RPE cells expressing HA-RILPL2 as in panel (B) were scored for TTBK2 at the centriole after serum starvation as in Fig S4. Shown are data from two independent experiments; >25 cells per experiment. No significant difference was detected. **(B, F)** RPE cells expressing HA-RILPL2 as in panel (B) were scored for endogenous MyoVa at the centriole; shown are data from two independent experiments; >50 cells per experiment for each condition. ****P < 0.0001 by *t* test.

verification of all plasmids was performed by Sequetech (http://www.sequetech.com). Myc-LRRK2 was obtained from Addgene (#25361) and the R1441G mutation introduced as described (Purlyte et al, 2018). N-terminally tagged GFP-RILPL2 was cloned into pcDNA5D using Gibson assembly, whereas HA-RILPL2 was cloned into BamH1 and Not1 sites of pcDNA5D by restriction cloning. Mouse MyoVa mCherry plasmid was a gift of Prof. Jim Spudich (Stanford University). MyoVa GTD-mCherry (1,421–1,880) and exon D (1,320–1,345) deletion mutants were made by site directed mutagenesis. For bacterial expression, MyoVa GTD, Rab10-Q63L (1–181), and Mst3 were kind gifts from Amir Khan (Trinity College, Dublin, Ireland). Rab8a Q63L was cloned into a pET14b expression plasmid

using Gibson assembly. mKO2-PACT was generated by amplifying the PACT domain of Pericentrin from cDNA of RPE cells as described (Sobu et al, 2021). To generate a plasmid-encoding Rab10 shRNA#1 and #2-resistant wild-type Rab10, inverse PCR was performed with primers (#1 F 5'-ACTGCAAATATGGGATACAGCAGG-3', #1 R 5'-TTAATCTTCTTTCCTTGTAATTCAACTG-3'; #2 F 5'-CACGGCATCAGG-TTTTTTGAGACTAG-3', R 5'-TTCTCTAGCGATCTGTTCTCCTTTTCC-3') myc-Rab10 plasmid was used as a template, and the PCR product was self-ligated with T4 DNA ligase, and T4 polynucleotide kinase before transformation. Point mutants in MyoVa mCherry and GTD-mCherry were introduced using PCR primers (R1528A - R 5'-TACTTTCTGAT-CATCGTTCAG, F 5'-GCATCATTGCTGACATCAACAATTAAC; R1755A/1757A F

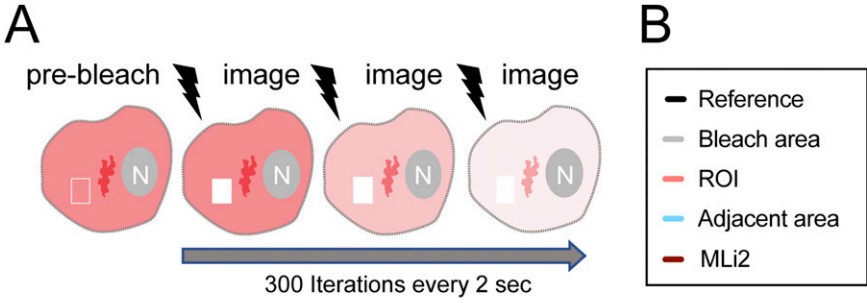

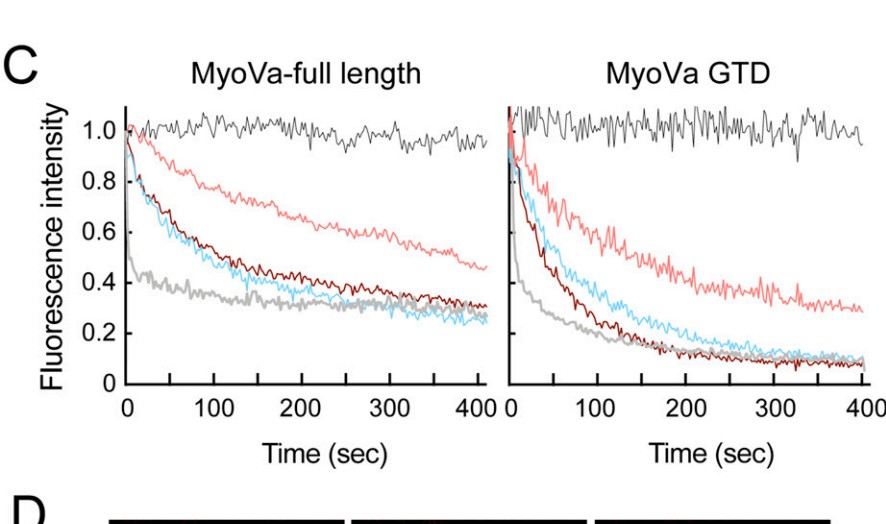

**Figure 9. Fluorescence loss in photobleaching analysis of MyoVa sequestration.**
**(A)** Schematic of fluorescence loss in photobleaching method. Cells are bleached 300 times over a region (~2% of cell area, usually 2 × 2 μm) distinct from the region of interest, once every (Sobu et al, 2020) *Preprint* 2 s. This will decrease the cytoplasmic fluorescence over the cell with time. Fluorescence over the bleached region and the region of interest is monitored continuously. **(B, C)** Legend for (C); (C) Relative fluorescence intensity as a function of time for the indicated regions. **(C, D)** Still micrographs from the experiment shown in (C) (Video 1). MyoVa full length-mCherry is shown in red; times (seconds) are indicated.

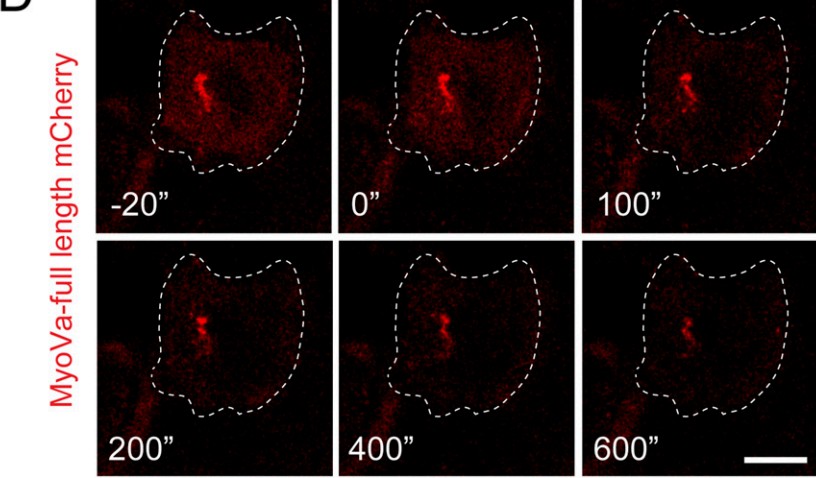

5′-GCTTTTGCAAGTGGCAAAGGCAACTGACGATGACGCAG, R 5′-GCGTCA-TCGTCAGTTGCCTTTGCCACTTGCAAAAGCTG). Base changes were confirmed by sequencing.

### Cell culture, transfections, and ciliogenesis

HEK293T, A549, and hTERT-RPE cells were obtained from American Type Culture Collection cultured in high-glucose DMEM with 10% fetal bovine serum. HEK293T cells were transfected with Polyethylenimine HCl MAX 4000 (PEI) (Polysciences, Inc.) as described (Reed et al, 2006). RPE cells were transfected with Fugene HD (Promega), whereas A549 cells were transfected using

Lipofectamine 3000 according to the manufacturer. RPE cells stably expressing mKO2-PACT were generated by lentivirus as described by Sobu et al (2021). Cells were checked routinely for mycoplasma using either MycoALert Mycoplasma Detection Kit (LT07-318; Lonza) or PCR. Cells were grown to confluence in DMEM with serum. hTERT-RPE cells were ciliated by overnight serum starvation in DMEM medium alone. For A549 cells, the cells were plated at 80% confluency and 24 h later, subjected to 2% serum for 48 h. To identify primary cilia, cells were stained for a cilia marker, Arl13b. For monitoring CP110 release, RPE cells were serum-starved for 1 h. Cultures of primary rat astrocytes were obtained by antibody panning from rat pups as described (Foo et al, 2011; Dhekne et al, 2018). In brief, 6–10 postnatal Sprague–Dawley

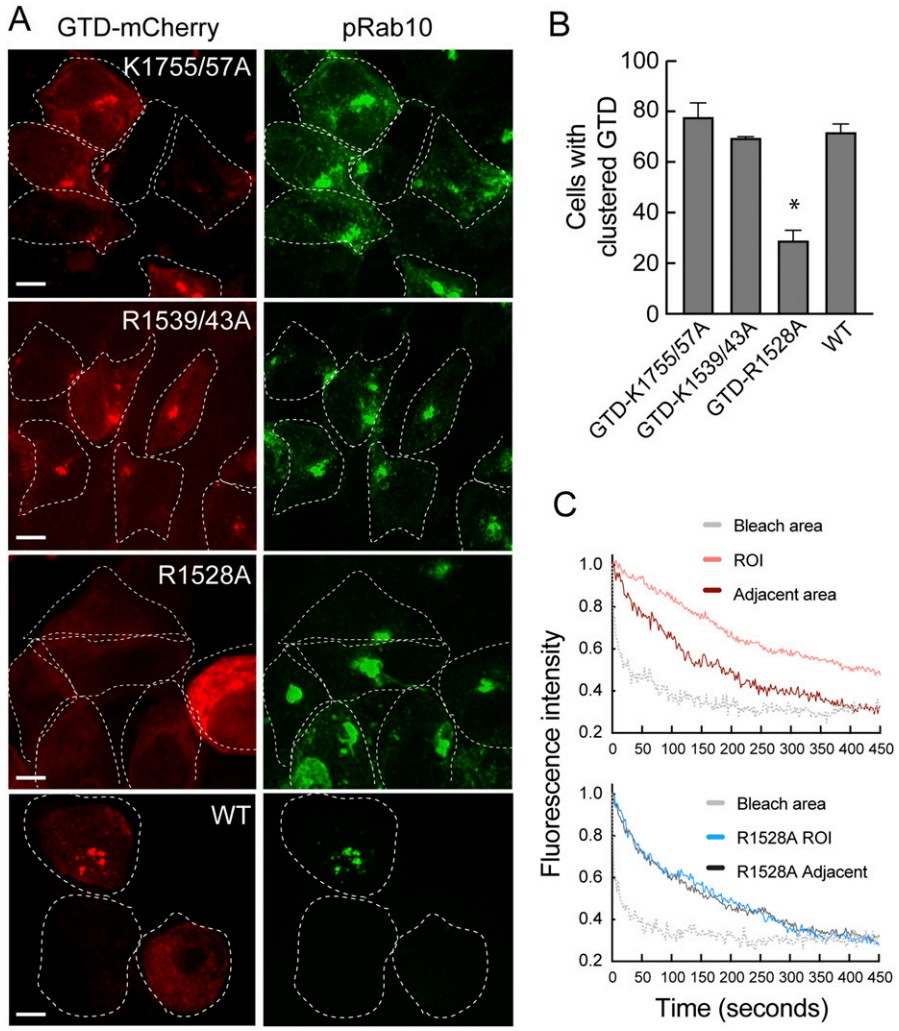

**Figure 10. PhosphoRab10 binding drives MyoVa sequestration.**
**(A)** Localization of MyoVa globular tail domain (GTD) mutants in cells expressing R1441G LRRK2, monitored by pRab10 staining. Cells were transfected with the indicated mutant GTD constructs for 24 h and stained as in Fig 2. **(B)** Quantitation of cells with clustered GTD for the indicated plasmids shown below. Shown are data from two independent experiments, >200 cells each. *P = 0.013. **(C)** Fluorescence loss in photobleaching analysis of cells expressing full length MyoVa or Full length MyoVa harboring the indicated mutation as in Fig 9.

rat cortices were mechanically and enzymatically dissociated to produce single cells. They were passed over successive, antibody-coated negative panning plates to rid the suspension of microglia, endothelial cells, and oligodendrocyte precursor cells before selecting for astrocytes with an anti-ITGB5–coated plate. Astrocytes that attached to the anti-ITGB5-coated plate were trypsinized and plated onto poly-D lysine coated ACLAR coverslips. The cells were incubated at 37°C with 10% $CO_2$ and 90% $O_2$ for 1–3 wk.

## Lentivirus production

Lentivirus-based shRNA was used for gene knockdown of human Rab10, human MyoVa, and mouse RILPL2. The lentiviral vector was co-transfected with packaging vectors psPAX2, pMD2 VSV-G in HEK293T cells using PEI. pLKO-puro-scramble was used as a control. After 48 h, culture supernatants were collected and virus concentrated 10× overnight with Lenti-X concentrator (Clontech) according to the manufacturer. Cells were transduced with virus using polybrene (2 μg/ml). To make stable cells, infected cells were selected with puromycin (1–2 μg/ml) 48 h after infection. Expression

of the target protein was verified by real time RT-PCR and/or immunoblotting.

## Gene knockdown

For silencing of the Rab10 gene, lentivirus pLKO.1-Rab10 shRNA was used to infect HEK293T cells as described (Dhekne et al, 2018). For human Rab10, pSIH vector was also used with a GFP reporter. The shRNA sequences for human Rab10 are as follows:

#1 5′-GATCCGAAGATCAAGCTACAGATCTTCCTGTCAGAATCTGTAGCTT-GATCTTCTTTTTG #2 5′-GATCCTTGCAAGGGAGCATGGTATTACTTCCTGT-CAGATAATACCATGCTCCCTTGCAATTTTTG

#3 5′-GATCCCAAGTGTGACATGGACGACAACTTCCTGTCAGATTGTCGT-CCATGTCACACTTGTTTTTG was cloned using EcoRI and BamHI restriction sites. For silencing human MyoVa in 293T cells, four shRNA (#KD1—TRCN0000382038, KD2—TRCN0000381587, KD3—TRCN0000381168, and #KD4—TRCN0000382222) were tested by Western blot. Because robust down-regulation was obtained using #KD1 and #KD2, which were used for further experiments. After 48 h, the cells were plated onto coverslips for ciliogenesis or processed to determine knockdown efficiency by qPCR. After 72 h, the cells were lysed for Western

blot analysis of the MyoVa knock-down. For RILPL2 knockdown in MEFs, two shRNA targeting mouse RILPL2 were obtained from Sigma-Aldrich (TRCN0000176476, TRCN0000200318). The RILPL2 knockdown in MEFs was verified by qPCR. Control shRNA sequence (cloned either into pSIH or pLKO) was as in Addgene Plasmid #1864 and used as control.

## Lysing and immunoblotting

Cells were lysed 24 h after transfection in ice-cold lysis buffer containing 50 mM Tris/HCl, pH 7.5, 150 mM NaCl, 1% (vol/vol) Nonidet P-40 (NP-40), 5 mM $MgCl_2$, 5 mM ATP, 1 mM EGTA, 1 mM sodium orthovanadate, 50 mM NaF, 10 mM 2-glycerophosphate, 5 mM sodium pyrophosphate, 0.1 $\mu$g/ml mycrocystin-LR (Enzo Life Sciences), 1 mM DTT, and EDTA-free protease inhibitor cocktail (Sigma-Aldrich). Lysates were centrifuged at 13,800$g$ for 15 min at 4°C and supernatants quantified by Bradford assay (Thermo Fisher Scientific) and 60 and 400 $\mu$g of proteins used for immunoblotting and co-immunoprecipitation assay, respectively. Antibodies were diluted in a blocking buffer containing 5% milk in 0.05% Tween in TBS and incubated overnight on the blots. Phos-tag (Fujifilm Wako) gels were made according to the manufacturer using 40 $\mu$M Phos-tag, 80 $\mu$M $MnCl_2$ in a 10% acrylamide-bis-acrylamide gel. The Phos-tag gel was washed for 15 min in 10 mM EDTA/Water, followed by 5-min wash in 1 mM EDTA/water and finally, distilled water. Phosphorylated proteins were transferred to nitrocellulose for 20 min using a Bio-Rad Transblot system and staining with antibodies as mentioned above.

## Real-time PCR analysis

Lysis for qPCR was performed by solubilizing cells in 500 $\mu$l Trizol. RNA was extracted according to the manufacturer. cDNA was synthesized from 500 ng total RNA using Expand Multiscribe cDNA synthesis kit (4368814) by Applied Biosystems. 1:10 diluted cDNA was then subjected to qPCR using Powerup Sybr green master mix (Thermo Fisher Scientific) in an Applied Biosystem ViiA 7 qPCR machine. qPCR primers for MyoVa were from Origene (hMyoVa Fw - 5′ CTCACACGAACTCCTGCAAA, hMyoVa Rv - 5′ AGGGGTAGTGGCATT-GAGTG), for RILPL2 from Sigma-Aldrich (mRILPL2 Rv - 5′ ATC-CTTCTCTCTTCTTCCTC, mRILPL2 Fw - 5′ AAGATGGTAGTGGATCTGAC). The housekeeping gene primers were mouse HPRT (mHPRT Rv - 5′ TTTACTGGCAACATCAACAG, mHPRT Fw - 5′ AGGGATTTGAATCACGTTTG), human GAPDH (hGAPDH fw - 5′ GGATTTGGTCGTATTGGG, hGAPDH Rv - 5′ GGAAGATGGTGATGGGATT). The mRNA expression analysis was performed using the delta delta Ct method.

## Co-immunoprecipitation

HEK293T cells expressing RILPL2-GFP, MyoVa mCherry, or myc-LRRK2 R1441G were harvested 24–48 h post-transfection and lysed. Equal amounts of extract protein were incubated with GFP-binding protein–conjugated agarose (ChromoTek) or anti-RFP antibody (Anti-RFP RTU; Rockland Biosciences)–conjugated protein G beads (Thermo Fisher Scientific) for 1 h at 4°C. Immobilized proteins were washed four times with 1 ml lysis buffer, eluted with 2× SDS loading buffer, and subjected to Bio-Rad Mini-PROTEIN TGX 4–20% gradient gels. After transfer to nitrocellulose membrane and antibody

incubation, the blots were visualized using the LI-COR Odyssey Imaging System.

## Generation of knockout A549 cells

Rab10 knock-out (Ito et al, 2016) cell lines have been described (Steger et al, 2017). The A549 RILPL1 knockout cell line has been described (Dhekne et al, 2018), and the RILPL2 knockout cell lines were made similarly using these guide pairs: sense A: GCCTG-TTGGGCCGCGAGCTTA; antisense A: GATGTCATACACGTCCTCGG and sense B: GTATGACATCTCCTACCTGTT, antisense B: GCCTCGGCGGT-CAGCTGGAAG.

## Generation of RILP2 knockout MEFs

RILPL2 knockout mouse line was originally acquired from The KOMP Repository and is now available from MMRRC.org (Stock number: 049453-UCD). Rilpl2$^{tm1a(KOMP)Wtsi}$ mice were initially bred with Taconic Total Body Cre mice expressing Cre recombinase (Model 12524) to produce the desired deletion. The mice were then bred and maintained on a C57Bl/6j background to remove the Cre recombinase allele. Genotyping of mice was performed by PCR using genomic DNA isolated from ear biopsies and the following primers, all at 10 pmol/$\mu$l: primer 1: 5′-AGTTCCGTGCCCTTTTATAGCTG-3′, primer 2: 5′-ACCACATGGCCTATTACCCAAACT-3′, and primer 3: 5′-ATAATAACCGGGCAGGGGGG-3′. PCR reactions were set up and run using KOD Hot Start Polymerase standard protocol. PCR bands were visualized on Qiaexcel (QIAGEN) using the standard DNA screening kit cartridge. The wild-type sequence leads to a 508-bp species, whereas the knockout produces a 697-bp species. Wild-type, het-erozygous, and homozygous RILPL2 knock-out MEFs were isolated from littermate-matched mouse embryos at day E12.5 generated from crosses between RILPL2 knockout heterozygous mice. The genotype of the cell was verified by PCR using genomic DNA as described above.

## Immunofluorescence staining and light microscopy

Cells were plated on collagen-coated coverslips transfected with indicated plasmids. Cells were fixed with 3% paraformaldehyde for 20 min, permeabilized for 3 min in 0.1% Triton X 100 (or 0.1% saponin for anti-pRab10 antibody staining), and blocked with 1% BSA in PBS. To stain centrioles with anti-CP110, anti-CEP164, or anti-EHD1, the cells were fixed with ice-cold 100% methanol for 5 min and gently washed with PBS followed by blocking with 1% BSA in PBS.

Antibodies were diluted as follows: mouse anti-Arl13b (1:2,000; Neuromab), rabbit anti-Arl13b (1:1,000; Proteintech); mouse anti-GFP (1:2,000; Neuromab); rabbit anti-RFP (1:1,000; Rockland Bio-sciences); mouse anti-Myc (1:2, 9E10 Hybridoma culture superna-tant, 1:1,000; BioLegend); rabbit anti pRab10 (1:1,000; Abcam); mouse anti-CEP164 (1:1,000; Santa Cruz); rabbit anti-CP110 (1:2,000; Pro-teintech); rabbit anti-EHD1 (1:1,000; Abcam), rabbit anti-LRRK2 (Clone UDD3; Abcam), rabbit anti-RILPL2 (1:500; Novus), chicken anti-rootletin (1:1,000; Aves), and mouse/rabbit anti-HA (1:1,000; Sigma-Aldrich). Anti-sera raised against RILPL2 in rabbit, which is described by Steger et al (2017), was used for staining immuno-panned primary rat astrocytes. Rabbit anti-MyoVa (1:1,000; Novus

Biologicals) was used for Western blots and immunofluorescence staining in 293T cells. Rabbit anti-MyoVa (1:1,000; Cell Signaling Technology) was used for Western blot in RPE cells. Goat anti-MyoVa (1:300, 17707; Santa Cruz) was used for staining primary rat astrocytes. Highly cross adsorbed H+L secondary antibodies (Life Technologies) conjugated to Alexa 488, Alexa 568, or Alexa 647 were used at 1:2,000. Nuclei were stained using 0.1 $\mu$g/ml DAPI (Sigma-Aldrich).

Images were obtained using a spinning disk confocal microscope (Yokogawa) with an electron multiplying charge coupled device camera (Andor) and a 100× 1.4 NA oil immersion objective. Images were analyzed using Fiji (https://fiji.sc/) and, as indicated, are presented as maximum intensity projections. Colocalization was quantified using the plugin JaCoP. Pearson's and Mander's correlation coefficients were used to quantify colocalization between fluorophores.

For time lapse imaging, $3 \times 10^4$ hTERT-RPE cells expressing mKO2-PACT were seeded into eight-well glass bottom dishes (Nunc Lab-Tek II Chambered Coverglass; Thermo Fisher Scientific) and cultured overnight. Cells were transfected with GFP-RILPL2, then after 24 h, the cells were serum-starved by replacing the media with Leibovitz's L-15 Medium, no phenol red (Thermo Fisher Scientific) without serum. Starting 15 min after starvation initiation, images were captured every 6 min using a spinning disk confocal with 63× 1.3 NA glycerol immersion objective at 37°C.

FLIP was performed on a Leica SP8 WLL laser scanning confocal microscope. MyoVa mCherry and GTD-mCherry cells were co-transfected with GFP-LRRK2 (R1441G) in HeLa (Kyoto) cells and imaged 16 h post-transfection. Cells were imaged at 256 × 256 resolution using bidirectional scanning and two-line averaging. A 1.5 × 1.5 $\mu$m square region ~7–8 $\mu$m distance from the perinuclear region was bleached for 600 ms (five iterations) using an Argon 488 laser. MyoVa mCherry (full length) images were captured with the standard photomultiplier tube using the 561 WLL laser (one iteration) every 2 s and the bleach–image cycle repeated 300 times (250 times for GTD-mCherry). Images were analyzed using the Fiji multi-measure tool on selected regions of interest. Adjacent non-ROI values are averaged from four different cytosolic areas of the cell. Data were plotted in GraphPad prism and $T_{1/2}$ were obtained by fitting one phase dissociation curves.

### Protein expression and purification

His$_6$ Rab10-Q63L (1–181), His$_6$ Mst3, His$_6$ Myosin Va GTD (1,464–1,855), and His$_6$ Rab8A Q63L were purified in *E. coli* BL21 (DE3 pLys) as described by Berndsen et al (2019) and Waschbüsch et al (2020). In brief, bacterial cells were grown at 37°C in Lucia Broth medium and induced at A$_{600}$ nm = 0.6–0.7 by the addition of 0.1 mM isopropyl-1-thio-$\beta$-D-galactopyranoside (Gold Biotechnology) and harvested after 18 h at 18°C. The cell pellets were resuspended in ice cold lysis buffer (50 mM Tris, pH 8.0, 10% [vol/vol] glycerol, 250 mM NaCl, 10 mM Imidazole, 5 mM MgCl$_2$, 0.5 mM DTT, 20 $\mu$M GTP, and EDTA-free protease inhibitor cocktail [Roche]), lysed by one passage through an Emulsiflex-C5 apparatus (Avestin) at 10,000 lbs/in$^2$, and centrifuged at 13,000 rpm for 25 min in a FiberLite F15 rotor (Thermo Fisher Scientific). Clarified lysate was incubated with cOmplete Ni-NTA resin (Roche) for 2 h at 4°C. Resin was washed three times with

five column volumes lysis buffer and eluted in 500 mM imidazole-containing lysis buffer. The eluate was buffer exchanged and further purified by gel filtration on Superdex-75 (GE Healthcare) with 50 mM Hepes, pH 7.5, 5% (vol/vol) glycerol, 150 mM NaCl, 5 mM MgCl$_2$, 0.1 mM tris(2-carboxyethyl)phosphine (TCEP), and 20 $\mu$M GTP.

### In vitro phosphorylation and microscale thermophoresis (MST)

His$_6$ Rab10 Q63L 1–181 was incubated with His$_6$-Mst3 at a molar ratio of 3:1 (substrate:kinase). The reaction buffer was 50 mM Hepes, pH 7.5, 5% (vol/vol) glycerol, 150 mM NaCl, 5 mM MgCl$_2$, 0.1 mM TCEP, 20 mM GTP, 6 $\mu$M BSA, 0.01% Tween-20, and 2 mM ATP (no ATP for negative control). The reaction mixture was incubated at 27°C for 30 min–4.5 h in a water bath. Phosphorylation completion was assessed by Western blot and PhosTag gel electrophoresis and found to saturate at 2 h. Immediately after phosphorylation, the samples were transferred to ice before binding determination.

Protein–protein interactions were monitored by MST using a Monolith NT.115 instrument (Nanotemper Technologies). His$_6$-MyoVa GTD was labeled using RED-NHS 2$^{nd}$ Generation (Amine Reactive) Protein Labeling Kit (Nanotemper Technologies). For all experiments, the unlabeled protein partner was titrated against a fixed concentration of the fluorescently labeled GTD (100 nM); 16 serially diluted titrations of the unlabeled protein partner were prepared to generate one complete binding isotherm. Binding was carried out in a reaction buffer in 0.5-ml Protein LoBind tubes (Eppendorf) and allowed to incubate in the dark for 30 min before loading into NT.115 premium treated capillaries (Nanotemper Technologies). A red LED at 10% excitation power (red filter, excitation 605–645 nm, emission 680–685 nm) and IR-laser power at 20% was used for 30 s followed by 1 s of cooling. Data analysis was performed with NTAffinityAnalysis software (NanoTemper Technologies) in which the binding isotherms were derived from the raw fluorescence data and then fitted with both NanoTemper software and GraphPad Prism to determine the $K_D$ using a non-linear regression method. The binding affinities determined by the two methods were similar. Shown are averaged curves of pRab10 and Rab10 from single readings from two different protein preparations.

### Statistics

Graphs were made using GraphPad Prism 6 software. Error bars indicate SEM. Unless otherwise specified, a *t* test was used to test significance. Two tailed *P*-values < 0.05 were considered statistically significant.

# Supplementary Information

# Acknowledgements

This research was funded by a grant to SR Pfeffer from the US National Institutes of Health (DK37332), and grants from the Michael J Fox Foundation

for Parkinson's research. We are grateful to Dr. Dario Alessi for critical input and enthusiastic support and Dr. Shahzad Khan for providing rat G2019S$^{+/-}$ astrocytes.

## Author Contributions

HS Dhekne: conceptualization, data curation, formal analysis, investigation, visualization, methodology, and writing—original draft, review, and editing.

I Yanatori: conceptualization, data curation, formal analysis, investigation, visualization, methodology, and writing—original draft, review, and editing.

EG Vides: conceptualization, data curation, formal analysis, investigation, visualization, methodology, and writing—review and editing.

Y Sobu: conceptualization, data curation, formal analysis, investigation, visualization, methodology, and writing—review and editing.

F Diez: resources.

F Tonelli: resources.

SR Pfeffer: conceptualization, data curation, formal analysis, supervision, funding acquisition, visualization, project administration, and writing—original draft, review, and editing

## Conflict of Interest Statement

The authors declare that they have no conflict of interest.

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
