## [Reviewer comments · Life Science Alliance]

Life Science Alliance

LRRK2-phosphorylated Rab10 sequesters Myosin Va with RILPL2 during ciliogenesis blockade

Herschel Dhekne, Izumi Yanatori, Edmundo Vides, Yuriko Sobu, Federico Diez, Francesca Tonelli, and Suzanne Pfeffer

DOI: <https://doi.org/10.26508/lsa.202101050>

Corresponding author(s): Suzanne Pfeffer, Stanford University

Review Timeline:

Submission Date:	2021-02-08
Editorial Decision:	2021-02-24
Revision Received:	2021-02-26
Accepted:	2021-03-01

Scientific Editor: Shachi Bhatt

Transaction Report:

Please note that the manuscript was previously reviewed at another journal and the reports were taken into account in the decision-making process at Life Science Alliance.

February 24, 2021

RE: Life Science Alliance Manuscript #LSA-2021-01050-T

Dr. Suzanne R. Pfeffer
Stanford University
Dept. of Biochemistry
Stanford University
School of Medicine
Stanford, 279 Campus Drive B400 94305-5307

Dear Dr. Pfeffer,

Thank you for submitting your revised manuscript entitled "LRRK2-phosphorylated Rab10 sequesters Myosin Va with RILPL2 during ciliogenesis blockade". We would be happy to publish your paper in Life Science Alliance pending final revisions necessary to meet our formatting guidelines.

For a brief overview, the manuscript was submitted and peer-reviewed at one of the Life Science Alliance (LSA) partner journals. The authors shared the manuscript and reviewers' comments with the LSA editors, who deemed the manuscript to have sufficient advance and quality to be published in Life Science Alliance.

Along with the points listed below, please also attend to the following,

- please add Keywords, Category, and Summary blurb/Alternate Abstract for your manuscript in our system
- please add Contributions of all Authors in our system
- please upload your main and supplementary figures as single files
- LSA allows supplementary figures, but not EV Figures; please update your callouts for the Supplementary Figures in the manuscript Fig EV1A = Fig S1A; while supplementary figures use the system supplementary Fig S1
- please add an explanation of Panel E in the legend for Figure EV1=S1
- please add callouts for Figures 4D; 7C and D to your main manuscript text
- please add scale bars for Figures 2D, 4B, 9D, EV2A, C, D, EV3B

Please also send in a point-by-point rebuttal to the remaining of the Reviewer 2's concerns.

To avoid unnecessary delays in the acceptance and publication of your paper, please read the

following information carefully.

A. FINAL FILES:

B. MANUSCRIPT ORGANIZATION AND FORMATTING:

Sincerely,

Shachi Bhatt, Ph.D.

Executive Editor

Life Science Alliance

<https://www.lsjournal.org/>

Interested in an editorial career? EMBO Solutions is hiring a Scientific Editor to join the international Life Science Alliance team. Find out more here -

https://www.embo.org/documents/jobs/Vacancy_Notice_Scientific_editor_LSA.pdf

Response to Reviewer Comments

Referee #2:

I am sorry that the revisions still have not made the paper interesting enough for me. My major concerns are:

1. The studies of cilia formation as the functional readout are very limited and preliminary and no real mechanism is pursued. The paper focuses on where proteins are in the cell and how they interact etc., but not on how they affect cellular function. Unless one can link the findings to functional consequences, they have limited interest for me.

A major discovery related to pathogenic LRRK2 function is that it completely switches the effector preferences of phosphorylated Rab GTPases. Thus, to understand the role of LRRK2, it is critical to identify phosphoRab binding partners and show what they do. Here we report for the first time, that MyoVa is a pRab10 specific binding protein and binds with submicromolar affinity, tighter than most Rab effector interactions. We show that simple LRRK2 generation of phosphoRab10, which we have shown previously is sufficient to block ciliogenesis (Dhekne et al., eLife 2018;7:e40202), completely redistributes MyoVa in cells, and holds it at the mother centriole for 5-10 minutes, as shown by fluorescence loss in photobleaching. MyoVa is already known to be critical for ciliogenesis and its retardation over the mother centriole will surely contribute to pRab10 cilia blockade. Unfortunately, RILPL1 also participates in cilia blockade so simple depletion of MyoVa would still yield a cilia defect due to the continued presence of pRab10-RILPL1 complexes. We are sorry that the reviewer didn't find this story interesting enough but for cell biologists, the relocation is completely unexpected, carefully and quantitatively documented and likely highly consequential.

2. The resistance to examining LRRK2 in neurons is surprising. I think that this is necessary for this study to have physiological and disease relevance.

We have shown previously in mutant mouse models (R1441C LRRK2) that cholinergic interneurons of the dorsal striatum show a cilia defect (eLife 2018;7:e40202 doi: [10.7554/eLife.40202](https://doi.org/10.7554/eLife.40202)) and more recently we have shown that astrocytes show a much more broad ciliation defect across the striatum in 3 LRRK2 mutant lines and PPM1H phosphatase KO animals (watch BioRxiv in the next days). LRRK2 is more abundant in astrocytes than neurons--the phenotype we observe is seen in multiple cell types and is not neuron-specific. This paper already includes 10 figures, 5 movies and 4 supplements. While neurons are surely interesting, the present story does not require additional study of neurons to support all of its conclusions. And even if dopaminergic neurons die in Parkinson's disease, this may have to do with LRRK2-driven loss of cilia in other cell types in the striatum, failing to provide GDNF neuroprotection.

3. Given that there is already a literature showing that mutant LRRK2 affects ciliogenesis and that this depends on Rab10 phosphorylation (as discussed in introduction) and RILP2 and Rab10 have been shown previously to bind Myosin Va, this paper does not add much to the existing literature. That is not true. This paper shows that RILPL2 is more likely to interact and be relocalized by pRab10 binding than by MyoVa binding. That they all bind is trying to tell us something very important. Rab10 doesn't bind to the brain MyoVa isoform and this is the first exploration of this tripartite connection--and it was not known previously that Rab phosphorylation relocalizes RILPL2 with MyoVa to the mother centriole.

4. Likewise, the paper does not show any dependence of RILPL2 for inhibition of cilia formation by mutant LRRK2. This obvious experiment is not reported (despite the suggestion in the previous round

of reviews) - maybe it was done and there was no effect, which gives me even greater cause for concern.

The reviewer appears to have entirely missed Figure 8 where this is documented and quantified. RILPL2 depletion or KO does not affect primary ciliogenesis in mouse embryonic fibroblasts, however, overexpressing RILPL2 in WT or KO MEFs dominantly blocks ciliogenesis and we show that this occurs without apparent blockage of TTBK2 centriolar recruitment but with a decrease in peri-centriolar MyoVa. RILPL2 depletion does not affect the distribution of MyoVa since MyoVa interacts directly with pRab10; RILPL2 is 10X less abundant than MyoVa and 240X less abundant than Rab10.

It is important to note that what a protein does in the absence of Rab10 phosphorylation may be quite different from what a protein does under conditions of Rab10 phosphorylation and inhibitory complex formation—and this is the more important distinction.

5. What happens if they express non-phosphorylatable Rab10 in Rab 10-depleted cells? - they need to find a way of testing the role of the phosphorylation - this was suggested by two reviewers previously and was not done.

We cannot do this--if the reviewer would have read the response to the reviews they would have learned that phospho-mimetic mutants are non-functional as we have carefully documented (Dhekne et al., 2018). Specifically: Wild type pRab8A binds RILPL1 but the TE “phosphomimetic” does not (while the real phospho protein binds tightly) and it and the TA mutant have a non-normal subcellular localization. Wild type pRab10 binds RILPL1 but the TE binds poorly (when again it should bind tightly) and the TA mutant is not membrane associated: Rab8 TE and Rab10 TA are not properly prenylated. None of the mutants can rescue ciliation phenotypes in knockout cells. Thus these are not possible experiments.

We are grateful to the editor of Life Science Alliance for agreeing to publish our story.

March 1, 2021

RE: Life Science Alliance Manuscript #LSA-2021-01050-TR

Dr. Suzanne R. Pfeffer
Stanford University
Dept. of Biochemistry
Stanford University
School of Medicine
Stanford, 279 Campus Drive B400 94305-5307

Dear Dr. Pfeffer,

Thank you for submitting your Research Article entitled "LRRK2-phosphorylated Rab10 sequesters Myosin Va with RILPL2 during ciliogenesis blockade". It is a pleasure to let you know that your manuscript is now accepted for publication in Life Science Alliance. Congratulations on this interesting work.

DISTRIBUTION OF MATERIALS:

Again, congratulations on a very nice paper. I hope you found the review process to be constructive and are pleased with how the manuscript was handled editorially. We look forward to future exciting submissions from your lab.

Sincerely,

Shachi Bhatt, Ph.D.

Executive Editor

Life Science Alliance

<https://www.lsjournal.org/>

Interested in an editorial career? EMBO Solutions is hiring a Scientific Editor to join the international Life Science Alliance team. Find out more here -

https://www.embo.org/documents/jobs/Vacancy_Notice_Scientific_editor_LSA.pdf